# Protective Effects of Mulberry (*Morus atropurpurea* Roxb.) Leaf Protein Hydrolysates and Their In Vitro Gastrointestinal Digests on AAPH-Induced Oxidative Stress in Human Erythrocytes

**DOI:** 10.3390/foods12183468

**Published:** 2023-09-18

**Authors:** Chongzhen Sun, Hongyan Li, Xiaodan Hui, Yurong Ma, Zhina Yin, Qingsong Chen, Cong Chen, Hui Wu, Xiyang Wu

**Affiliations:** 1School of Public Health, Guangdong Pharmaceutical University, Jianghai Avenue 283, Haizhu District, Guangzhou 510006, China; sunchongzhen@gdpu.edu.cn (C.S.); 2112241029@gdpu.edu.cn (H.L.); yinzhina@gdpu.edu.cn (Z.Y.); qingsongchen@aliyun.com (Q.C.); 2Guangdong Provincial Key Laboratory of Tropical Disease Research, School of Public Health, Southern Medical University, Guangzhou 510515, China; xiaodan.hui@lincolnuni.ac.nz; 3Shenzhen Key Laboratory of Food Nutrition and Health, Institute for Advanced Study, Shenzhen University, Shenzhen 518060, China; yurongm@163.com; 4Department of Food Science and Engineering, Jinan University, Huangpu Road 601, Guangzhou 510632, China; 18520392111@163.com; 5College of Food Science and Engineering, South China University of Technology, Guangzhou 510640, China

**Keywords:** mulberry leaf, digested protein hydrolysates, erythrocytes, oxidative stress, natural antioxidants

## Abstract

Mulberry leaf protein hydrolysates (HMP), and their in vitro gastrointestinal digests (GHMP), have shown favorable chemical antioxidant activities. The aim of this study is to investigate the potential protective effects of HMP and GHMP against 2,2′-azobis(2-amidinopropane) dihydrochloride (AAPH)-induced oxidative stress in human erythrocytes. The inhibition rate of hemolysis, the reactive oxygen species (ROS) level, the concentration of malondialdehyde (MDA), the reduced glutathione (GSH) and oxidized glutathione (GSSH), and the enzymatic activities of total superoxide dismutase (SOD), catalase (CAT), and cellular glutathione peroxidase (GSH-Px) were evaluated as the biomarkers of oxidative status in human erythrocytes. The results showed that HMP and GHMP effectively inhibit the occurrence of erythrocyte hemolysis in the range of 0.025–1.0 mg/mL, and the inhibition rates of HMP and GHMP reached 92% and 90% at concentrations of 0.4 mg/mL and 1.0 mg/mL, respectively. HMP and GHMP reduced the AAPH-induced oxidative hemolysis damage via suppressing the generation of ROS by inhibiting the formation of MDA, maintaining the balance of GSH/GSSG, and preserving the activities of the antioxidant enzymes, including SOD, GSH-Px, and CAT. Our findings revealed that both HMP and GHMP could be used as natural antioxidants, and have the potential for further application in the development of functional foods.

## 1. Introduction

Reactive oxygen species (ROS), the by-products of normal cellular metabolism, are highly reactive non-specific molecules derived from oxygen metabolism, such as oxygen free radicals [1]. Under normal physiological conditions, the balance between the production and neutralization of ROS is usually favorable in the presence of regulatory mediators (antioxidants and enzymes) in signaling processes [2]. However, if ROS overwhelm the regulatory capacity of the body, a state of oxidative stress will occur. In this case, biological oxidative damage can be provoked on the relevant molecules such as lipids, proteins, and nucleic acids [1]. Hence, the application of exogenous antioxidants is necessary to maintain proper physiological function for offsetting this oxidative stress. Due to the potential toxicity of synthetic antioxidants, screening for effective and nontoxic natural antioxidants, in particular those derived from medicinal and dietary plants, has been a global trend [3].

Bioactive antioxidant peptides have attracted attention from food manufacturers and scientists in recent years, due to their functionalities and their potential for preventing chronic diseases related to oxidative stress, such as cancers and type 2 diabetes [4]. Natural peptides isolated and purified from medicinal and dietary plants are found to have relatively stronger bioactivities and stability, and easier absorption than synthetic antioxidants and proteins [5,6]. Mulberry leaf, as a medicine and food plant, is widely distributed in Asian countries. It is mainly used for silkworm breeding, with low utilization rate and huge waste [7]. In addition, mulberry leaves are always consumed as functional foods and additives, such as mulberry leaf noodles, soup, and drink powders while, in Western countries, mulberry leaf is fermented into mulberry wine for consumption [8]. To date, previous studies in mulberry leaf have mainly focused on its bioactive compounds, such as flavonoids, phenolic acids, alkaloids, and polysaccharides [9,10]. In addition to these bioactive compounds, mulberry leaf is abundant in protein, with dry leaves containing 17–25% protein [11,12,13]. Our previous study [14] revealed that mulberry protein hydrolysates (HMP), as well as its gastric digesta, GHMP, could be used as natural antioxidants based on chemical methodologies (DPPH and ABTS scavenging abilities). Three novel antioxidant peptides were also purified from HMP [8].

Owing to the better correlation with biological systems compared to the chemical methods, in vitro cell-based studies are always conducted to evaluate the antioxidant ability against oxidative stress and elucidate the underlying mechanisms of oxidative stress. Human erythrocytes are one of the most specialized cells, and also represent the most abundant cells in the body [15]. Their unique structural characteristic is constituted by the absence of nuclei and mitochondria. The main function of the erythrocytes is to produce the co-enzyme factors to maintain the balance of osmotic pressure and resist oxidative stress. When the erythrocytes are employed to evaluate the protective effects of antioxidants on oxidative damage, it must be initially stimulated to oxidative damage [16]. 2,2-Azobis(2-methylpropionamide)-dihydrochloride (AAPH) is a commonly used azo-based free radical initiator in cell models [17]. The stimulation of AAPH can cause excess ROS in erythrocytes. ROS can react with phospholipid unsaturated fatty acids in the cell membrane to generate lipid peroxides, leading to cell membrane damage, and thereby inducing the hemolysis of erythrocytes. The oxidized radicals can be detoxified by antioxidant enzymes in erythrocytes, including superoxide dismutase (SOD), catalase (CAT), and glutathione peroxidase (GSH-Px), to avoid oxidative stress [18]. Therefore, AAPH-induced hemolysis in erythrocytes can directly reflect the biologically related free-radical-scavenging ability by measuring the content of ROS and malondialdehyde (MDA), as well as the activities of intracellular antioxidant enzymes [19].

Our previous study [14] illustrated that HMP and GHMP could effectively inhibit the hemolysis of sheep red blood cells (RBCs). However, the potential mechanism of inhibiting hemolysis of erythrocytes by HMP and GHMP has not been studied yet. Herein, this study aims to elucidate the antioxidant mechanisms of HMP and GHMP at the cellular level by further validating the inhibitory effects of HMP and GHMP on AAPH-induced erythrocyte hemolysis using human erythrocytes. The findings of this study will contribute to a better understanding of the antioxidant properties of mulberry leaf protein hydrolysate and suggest that HMP and GHMP have the potential to be utilized as natural antioxidants in the development of functional foods.

## 2. Materials and Methods

### 2.1. Materials

Mulberry leaves (*Morus atropurpurea* Roxb.) were provided by the Institute of Sericulture and Agricultural Products Processing (Guangzhou, China). Neutrase (0.8 AU/g) was purchased from Novozymes (Bagsvaerd, Denmark). 2,2′-azobis (2-amidinopropane) dihydrochloride (AAPH) and 2′,7′-dichlorofluoresceindiacetate (DCFH-DA) were purchased from Sigma Company (St. Louis, MO, USA). Assay kits for the determination of total protein (BCA kit), malondialdehyde (MDA, A003-2), total superoxide dismutase (SOD, A001-3), glutathione peroxidase (GSH-Px, A005), reactive oxygen species (ROS, E004), catalase (CAT, A007-1), and total glutathione/oxidative glutathione (T-GSH/GSSG, A061-1) were bought from Nanjing Jiancheng Co., (Nanjing, China). The ultrapure water was purchased from Watsons (Guangzhou, Guangdong, China). Phosphate-buffered saline solution (PBS, pH = 7.4) was purchased from Thermo Fisher Scientific Inc. (Waltham, MA, USA). Human blood samples were obtained from the second affiliated hospital of Guangzhou University of Traditional Chinese Medicine. Other chemicals and reagents in this study were of analytical grade (Aladdin Co., Shanghai, China).

### 2.2. Preparation of HMP

HMP was prepared using the methods described in our previous studies [14,20,21] and was further purified by Macroporous adsorption resin DA201-C analysis [22]. Additionally, the peptide content of HMP was determined based on the o-phthaldialdehyde method [23], the peptide content was 189.57 mg/g (serine was used as the standard).

### 2.3. In Vitro Gastrointestinal Digestion of HMP

The in vitro gastrointestinal digestion of HMP was performed according to our previous research [14]. After digestion and centrifugation, the supernatant was desalted using a dialysis bag (Mw cut-off was 100 Da). After desalting, the sample was pre-frozen at −80 °C for 12 h and then freeze-dried with a vacuum freeze dryer (SCIENTZ-18N, Ningbo, China) for 36 h to obtain GHMP.

### 2.4. Assay for Erythrocyte Hemolysis Induced by AAPH

The protective effect of HMP and GHMP against AAPH-induced erythrocytes hemolysis was measured according to our previous method [14]. RBCs were separated from plasma by centrifugation (1200× *g*, 5 min, 4 °C). The resulting erythrocytes were washed with PBS (pH = 7.4), and made into 20% erythrocytes suspension with PBS. Afterwards, the suspension was mixed with HMP or GHMP solution at the final concentrations of 0, 0.025, 0.125, 0.25, 0.4, 0.7, and 1.0 mg/mL. After incubation at 37 °C for 30 min, 100 mmol/L AAPH was added. The mixture was further incubated at 37 °C for 2 h, and then the mixture was diluted with PBS and centrifuged (1200× *g*, 10 min). The absorbance of the supernatant was measured at 540 nm (A). The complete hemolysis was obtained from adding ultrapure water into the suspension (B). The hemolysis rate was calculated as A/B × 100%. The remaining precipitated RBCs were used for subsequent analysis.

### 2.5. Morphological Changes in Erythrocytes Determined Using Scanning Electron Microscopy (SEM)

The treated erythrocytes were collected for morphological analysis. Imaging of erythrocytes was conducted using a scanning electron microscope (SEM) (Hitachi TM3000, Hitachi Ltd., Tokyo, Japan) [24].

### 2.6. Determination of Intracellular ROS Generation

The relative intracellular ROS level was detected using the ROS assay kit [25]. The precipitated RBCs were suspended with five volumes of PBS (pH = 7.4). The re-suspended erythrocytes were divided into A and B groups. Group A was used for determining the ROS level. An amount of 100 μL of RBCs suspension was centrifuged (1200× *g*, 8 min, 4 °C), the supernatant was discarded, and then the indicator DCFH-DA (final concentration of 10 μmol/L) was added to the erythrocyte suspension, with subsequent incubation at 37 °C for 30 min. Afterwards, the mixture was washed with PBS to completely remove the DCFH-DA outside the erythrocytes. At the end of washing, the RBCs containing a fluorescent probe were re-suspended with 600 μL of PBS. The fluorescence intensity was measured using a fluorescence microplate reader (excitation/emission, 485/535 nm). The results were reported as the percentage of DCF fluorescence intensity of control.

### 2.7. Measurement of the Antioxidant Capacity of Erythrocytes

The treated RBC suspension in group B was centrifuged at 1200× *g*, 4 °C for 8 min. The supernatant was discarded, and the remaining erythrocytes were completely lysed by five volumes of distilled water. The mixture was centrifuged again at 1200× *g*, 4 °C for 5 min, and then the supernatant was collected for the determination of protein, MDA, GSH-Px, GSH, GSSG, SOD, and CAT. The protein content was determined using a BCA assay kit. MDA content was measured using the thiobarbituric acid (TBA) method; the result was expressed as nanomole MDA nmol/mg protein. The contents of GSH and GSSG were measured using a GSH and GSSG assay kit, and the results were expressed as nanomole GSH/GSSG nmol/mg protein. The activities of SOD and GSH-Px were measured using a total superoxide dismutase assay kit with nitro-blue tetrazolium (NBT) and a cellular glutathione peroxidase assay kit, respectively. SOD was expressed as U/mg protein, and GSH-Px was reported as mU/mg protein.

### 2.8. Statistical Analysis

The data are presented as the mean ± standard deviation (SD, n = 3). Statistical calculations were performed by one-way analysis of variance using SPSS Statistics V17.0 (IBM, Armonk, NY, USA). Duncan’s multiple range tests were used to identify significant differences (*p* < 0.05) among treatment means. Origin Program 8.6 was used to draw diagrams.

## 3. Results and Discussion

### 3.1. The Inhibitory Effects of HMP and GHMP against AAPH-Induced Erythrocyte Hemolysis

Erythrocyte is normally used as a cell model to determine if one sample is resistant to oxidative stress, as the ability of erythrocytes in scavenging free radicals and ROS represents the direct antioxidant protection of the body [26]. AAPH is an azo-based free radical initiator, which can decompose at 37 °C to produce hydroperoxyl radicals once encountering the oxygen [17]. These hydroperoxyl radicals attack the cell membrane, and interact with phospholipids to cause lipid peroxidation, which can destroy the integrity of the cell membrane, leading to the hemolysis of RBCs. Despite the weaker DPPH and ABTS quenching ability of GHMP, both mulberry leaf protein (MP) and HMP have been proven to be natural antioxidants in our previous study [14]. One can further assure the intracellular antioxidant capacity of HMP and GHMP through determining the inhibitory effects of HMP and GHMP on AAPH-induced hemolysis in human RBCs. As shown in Figure 1, RBCs were incubated in PBS alone; they were very stable with limited hemolysis observed. After AAPH treatment, the RBCs were oxidatively damaged, and the hemolysis inhibition rate was as low as 52.32%. In contrast, the RBCs were pre-treated with HMP or GHMP with varying concentrations from 0.025 to 1 mg/mL, and the hemolysis inhibition rate of RBCs was significantly increased (*p* < 0.05). When the concentration of HMP and GHMP was added at 0.4 and 1 mg/mL, respectively, the inhibition rate reached the highest, and showed no obvious difference from the normal control group (*p* > 0.05). Our previous study has reported that both HMP and GHMP quenched the ABTS^+^, DPPH, and O_2_^-^ radicals by transferring a proton to electron-deficient radicals [8,14]. Herein, it was speculated that HMP and GHMP interacted with membrane lipid bilayers of RBCs, and terminated the radical chain reaction through proton donation, consequently protecting erythrocytes from AAPH-induced hemolysis. In addition, peptides can be absorbed by human cells where they act as antioxidants [8,27]. HMP and GHMP were mainly composed of polypeptides (0.5–6.5 kDa) and oligopeptides (<0.5 kDa), respectively [14]. These peptides could easily pass through the cell membrane to scavenge the intracellular free radicals, thereby inhibiting the hemolysis. When the RBCs were treated with 1 mg/mL HMP or GHMP without AAPH, the inhibition rate was nearly equal to the normal group (*p* > 0.05), suggesting that HMP and GHMP had no toxicity towards RBCs. In the range of 0.025–0.7 mg/mL, the hemolysis inhibition rate of the HMP group was significantly higher than that of GHMP at the same concentration (*p* > 0.05), which was in accordance with our previous findings [14].

### 3.2. The Protective Effects of HMP and GHMP on the Morphology of AAPH-Induced Erythrocyte Hemolysis

To characterize the erythrocyte damage induced by AAPH and the protection of HMP and GHMP, morphologic changes in erythrocytes were observed using SEM (Figure 2). The normal erythrocytes, as the control group, are shown in Figure 2A, obviously appearing to display the tridimensionality state with a typical biconcave shape and smooth surfaces and edges. There was no spike-like process extending out from the surface of RBCs. However, the exposure of AAPH significantly caused the structure failure and cell collapse of RBCs. The cell surface of AAPH-induced RBCs became rougher. Significant amounts of spike-like processes were observed to be extended out from the surface of RBCs (Figure 2B). When these RBCs were treated with 1 mg/mL of either HMP or GHMP, the perturbing effects caused by AAPH were neutralized. The amounts of spike-like processes were significantly reduced (Figure 2C,D, respectively) when compared to the AAPH-induced group, and the three-dimensional structure was restored. This observation is consistent with the results of the hemolysis inhibition ability of HMP and GHMP. Collectively, the antioxidant capacity of HMP and GHMP, against the oxidative damage of RBCs induced by AAPH, is clearly demonstrated.

### 3.3. HMP and GHMP Released AAPH-Induced Oxidative Stress via Reducing Intracellular ROS Accumulation

ROS usually participate in a variety of biological processes. However, once the balance between ROS generation and antioxidant defenses is unfavorable, the ROS accumulation may cause severe cell membrane and DNA damages, leading to oxidative stress, such as lipid peroxidation [28]. As a fluorescent probe for the determination of the relative levels of ROS, the non-polar compound, DCFH-DA, can cross over the cellular membrane and enter the cytoplasm [29]. DCFH-DA can be deacetylated by intracellular esterases to generate a polar, non-fluorescent compound, dichlorofluorescein (DCFH). DCFH can be oxidized to the fluorescent 2′,7′-dichlorofluorescein (DCF) by intracellular ROS derived from the metabolism of oxygen. The fluorescence intensity of cells would increase with the increasing level of intracellular free radicals. Therefore, the determination of the intracellular AAPH-induced ROS generation could reveal the corresponding antioxidant rationale of either HMP or GHMP in the inhibition of erythrocyte hemolysis. The AAPH-induced ROS generation and the inhibitory effect of HMP and GHMP are displayed in Figure 3. According to the figure, the stimulation of AAPH significantly increased by 8-fold the ROS levels in RBCs when compared to the blank group (*p* < 0.05). Pre-treatment with HMP or GHMP at varying concentrations from 0.025 to 1 mg/mL effectively inhibited these increased ROS levels in RBCs induced by AAPH. This inhibitory effect was in a concentration-dependent manner. The treatment with 0.7 mg/mL HMP and 1 mg/mL GHMP in RBCs reduced ROS levels by 66.67% and 55.57% compared to the AAPH-induced alone group, respectively (*p* < 0.05), and this had no obvious difference from the control group (treatment with HMP or GHMP alone). In addition, the fluorescence intensity of the HMP-treated group was lower than that of the corresponding GHMP-treated group (*p* < 0.05), suggesting that HMP had a greater inhibitory effect on reducing intracellular ROS generation than GHMP. This result is also in agreement with the results of the protective effects of HMP and GHMP against hemolysis and morphologic changes. Taken together, these findings significantly indicated that either HMP or GHMP suppressed the AAPH-induced oxidative stress in erythrocytes by inhibiting cellular ROS generation, thus inhibiting the hemolysis in RBCs.

### 3.4. HMP and GHMP Reduced the AAPH-Induced MDA Accumulation

Excess free radicals can damage cells by attacking the cell membranes, leading to lipid peroxidation. MDA refers to a degradation product from lipid peroxidation and is widely assessed as the marker of oxidative stress induced by ROS [30]. When MDA accumulates to a certain extent, it can destroy the structure of the cell membrane, thus leading to cellular metabolic disorder and cell apoptosis. Hence, an assessment of MDA level is beneficial for revealing the corresponding antioxidant rationale of antioxidants in inhibiting erythrocyte hemolysis. Herein, the effects of HMP and GHMP on AAPH-induced MDA accumulation in RBCs were determined, as shown in Figure 4. The treatment with either HMP or GHMP in RBCs did not cause significant MDA accumulation when compared to the blank group (*p* > 0.05), suggesting that HMP and GHMP caused no damage to the RBCs’ membrane. However, the stimulation of AAPH dramatically increased by 6-fold the MDA level compared to the control group (*p* < 0.05), whilst the MDA levels of both the HMP and GHMP groups were significantly lower than that of the AAPH-induced group (*p* < 0.05). These results suggested that, within the concentration from 0.025 to 1.0 mg/mL, HMP and GHMP effectively scavenged the free radicals induced by AAPH, and prevented lipid peroxidation, thus inhibiting the formation of MDA, which could protect the cell membrane from oxidative damage. This protective ability was in a concentration-dependent manner.

The MDA level in the 0.025, 0.125, and 0.25 mg/mL HMP-treated group was significantly lower than that in the corresponding GHMP-treated group (*p* < 0.05). When treated with 0.4, 0.7, and 1 mg/mL HMP or GHMP in RBCs, no obvious difference in MDA levels was found between these two groups (*p* > 0.05). This finding suggested that the inhibitory ability of HMP against lipid peroxidation was weakened after experiencing in vitro gastric digestion. This is highly consistent with the results of effective hemolytic inhibition rate and the strong ROS inhibitory ability of HMP.

### 3.5. HMP and GHMP Reversed the AAPH-Induced Reduction in the Ratio of GSH/GSSG in Erythrocytes

The defense system of the human body includes the enzymatic and non-enzymatic antioxidant systems. GSH and oxidized glutathione (GSSG) are the representatives of the non-enzymatic antioxidant defense systems [31]. GSH is considered to be one of the most important scavengers of ROS, which provides the first defense barrier via either scavenging free radicals or being the substrate of GSH-Px in the detoxification process of lipid peroxides, hydrogen peroxides, and electrophilic substances [32]. Hence, one of the ways to prevent oxidative damage is to reduce the consumption of GSH. GSH is extremely sensitive to oxidative damage caused by AAPH. Once the oxidative stress occurs, GSH would be consumed due to the direct attack of free radicals and the other repair processes requiring GSH participation. As shown in Figure 5A, compared with the blank group, when the RBCs were stimulated with AAPH, the content of GSH decreased significantly from 9.85 to 1.81 nmol/mg protein (*p* < 0.05). However, this effect was remarkably reversed by pre-treating with HMP and GHMP in AAPH-induced RBCs, in a concentration-dependent manner. When treated with 0.4, 0.7, and 1 mg/mL HMP or 0.7 and 1 mg/mL GHMP in RBCs, no significant difference in GSH content was found in comparison to the control group (*p* > 0.05). According to our previous study, it can be speculated that both HMP and GHMP reversed the loss of GSH, probably due to the abilities of scavenging free radicals and the inhibiting of ROS generation. There was no statistical difference between the HMP and GHMP groups, apart from the concentration of 0.4 mg/mL, in which the GSH content in the HMP (9.81 nmol/mg protein) group was significantly higher than the corresponding GHMP (8.13 nmol/mg protein) group (*p <* 0.05).

Under the conditions of oxidative stress, GSH can be oxidized to GSSG by GSH-Px. Excess GSSG can be reduced to GSH by nicotinamide adenine dinucleotide phosphate (NADPH) via the action of glutathione reductase [33]. The effect of HMP and GHMP on the GSSG content in RBCs was investigated, as shown in Figure 5B. When the erythrocytes were stimulated with AAPH, the content of GSSG in the erythrocytes significantly increased from 2.46 nmol/mg protein (PBS control group) to 8.20 nmol/mg protein (*p* < 0.05). However, this effect was considerably reversed by pre-treating with HMP and GHMP, with varying concentrations from 0.025 to 1 mg/mL, which is opposite to the effect on the GSH content. This finding indicated that HMP and GHMP could effectively inhibit the loss of GSH in RBCs, reduce GSSG generation, and thus maintain the balance of GSH and GSSG in RBCs. There was no obvious difference between the HMP and GHMP groups, in addition to the concentration of 1 mg/mL, in which the GSSG content in the HMP (2.51 nmol/mg protein) group was significantly lower than the corresponding GHMP (2.64 nmol/mg protein) group (*p* < 0.05).

The glutathione redox couple, GSH/GSSG, has a great importance in cells, which can regulate and maintain the appropriate cellular redox status [34]. Changes in the ratio of GSH/GSSG are fundamental in the fine-tuning of signal transduction, even under the mild oxidative stress that underlies physiological events such as cell cycle regulation and other cellular processes. Therefore, the ratio of GSH/GSSG can be used as a marker of oxidative stress. As shown in Figure 5C, stimulation with 100 mmol/L AAPH dramatically decreased the ratio of GSH/GSSG from 3.96 to 0.23 (*p* < 0.05) in RBCs, while with pre-treatment of HMP or GHMP with varying concentrations between 0.025 and 1 mg/mL, the ratio of GSH/GSSG in RBCs gradually increased in a concentration-dependent manner. In particular, when the concentration of the HMP or GHMP reached 1 mg/mL, the ratio of GSH/GSSG was close to that in the PBS control group, suggesting that 1 mg/mL HMP or GHMP could effectively inhibit the oxidative stress of RBCs by maintaining the balance of GSH and GSSG.

### 3.6. HMP and GHMP Prevented AAPH-Induced Changes in Activities of Antioxidant Enzymes in Erythrocytes

In addition to non-enzymatic antioxidant systems, there are some antioxidant enzymes in RBCs, such as SOD, GSH-Px, and CAT, that cooperate to resist oxidative stress and protect the body from free radical attacks [3]. Under normal physiological conditions, these antioxidant enzymes (SOD, GSH-Px, and CAT) can catalyze the conversion of ROS into less reactive species, forming the first barrier of defense against oxidative stress. Once oxidative stress occurs, the body will activate its defense system to remove excessive ROS to maintain the balance of the redox reaction and reduce oxidative damage. Among these enzymes, SOD mainly scavenges excess superoxide anion free radicals in cells to produce H_2_O_2_, while GSH-Px and CAT remove intracellular and extracellular hydrogen peroxide by reducing H_2_O_2_ to H_2_O [35]. Therefore, the changes in the activities of these enzymes can be used as the biomarkers of cell antioxidant response.

As can be observed from Figure 5, the activities of SOD (Figure 5D), GSH-Px (Figure 5E), and CAT (Figure 5F) in RBCs were significantly increased after stimulation with AAPH (*p* < 0.05), which were 2.33, 2.39, and 1.12 times higher than those in the PBS control group, respectively. AAPH stimulated the RBCs to produce free radicals, which triggered the activation of the enzymes defense system in RBCs, and thus resulted in the increased demand for these enzymes. With pre-treatment with HMP and GHMP (from 0.025 to 1 mg/mL), the demand for the SOD and GSH-Px were significantly reduced, in a concentration-dependent manner, compared to the AAPH-induced group (*p* < 0.05). In particular, 1.0 mg/mL HMP and GHMP exhibited the least demand for SOD enzyme activity compared to other concentrations. In addition, pre-treatment with 0.4, 0.7, and 1 mg/mL HMP displayed less demand for GSH-Px than that of GHMP (*p* < 0.05). This result suggested that in vitro digestion reduced the protective ability of HMP, increasing the demand for GSH-Px in RBCs. These findings indicated that treatment with HMP and GHMP could effectively alleviate the increased demand for SOD and GSH-Px in erythrocytes under oxidative stress. This is mainly due to the great free radical scavenging and ROS inhibitory abilities of HMP and GHMP.

Different from GSH-Px and SOD, CAT is not very sensitive to oxidative damage caused by AAPH. As shown in Figure 5F, CAT activity in the AAPH-induced group increased from 43.67 to 48.43 U/mg protein when compared to the PBS group (*p* < 0.05). Treatment with HMP or GHMP concentrations from 0.025 to 0.7 mg/mL showed no significant difference in CAT activity from the AAPH-induced group (*p* > 0.05). Among all the concentrations, only 1 mg/mL HMP effectively preserved CAT activity. This may be due to the relatively limited hydrogen peroxide produced by AAPH during the process of triggering oxidative stress. Meanwhile, GSH-Px would compete with CAT for H_2_O_2_ as the substrate. Herein, according to the results of GSH-Px and CAT activities, GSH-Px had a stronger competition than CAT, which may result in there being no significant difference in CAT activity between these concentrations.

### 3.7. Possible Antioxidant Mechanisms of HMP and GHMP Attenuate AAPH-Induced Oxidative Damage

The possible intracellular antioxidant mechanism of HMP and GHMP is illustrated in Figure 6. As a free radical promoter, AAPH would produce large amounts of ROS, such as O_2_^−^ and H_2_O_2_, when using it to stimulate RBCs. Excess ROS caused lipid peroxidation, producing a large amount of MDA. The destruction of the phospholipid bilayer caused damage to the cell membrane, ultimately leading to the hemolysis of the RBCs. Enzymatic antioxidant defense systems exist in the RBCs, such as SOD, CAT, and GSH-Px. SOD catalyzes the disproportionation reaction of O_2_^−^ to produce H_2_O_2_ with lower toxicity, while H_2_O_2_ inherent in the cell, and catalyzed by SOD, can be transformed into H_2_O with the participation of CAT. It can also react with GSH to generate H_2_O under the action of GSH-Px. In addition to the enzymatic antioxidant system, as the representatives of the non-enzymatic defense system in RBCs, GSH can be oxidized to GSSG under the action of GSH-Px, and then excessive GSSG can be reduced to GSH by NADPH. The enzymatic and non-enzymatic reactions maintain a dynamic balance between intracellular free radical production and scavenging by directly or indirectly participating in intracellular redox reactions, thus maintaining normal erythrocyte life activities. However, when erythrocytes are attacked by free radicals for a prolonged period of time, the intracellular defense system is insufficient in scavenging excess free radicals, resulting in the destruction of erythrocytes. With the addition of HMP and GHMP, ROS production was inhibited and free radicals were scavenged, thus controlling the oxidative damage caused by free radicals and maintaining the balance between enzymatic and non-enzymatic antioxidants. Overall, HMP and GHMP could attenuate AAPH-induced oxidative stress by inhibiting cellular ROS generation, inhibiting MDA production, maintaining GSH/GSSG balance, and preserving the activities of antioxidant enzymes, such as SOD, GSH-Px, and CAT. In addition to the intracellular antioxidant mechanism, there may also be some extracellular mechanisms, because our previous studies have shown that both HMP and GHMP can effectively remove DPPH, ABTS^+^, and O_2_^-^ free radicals [8,14]. In this research, HMP and GHMP were added before APPH, so it is likely that part of the free radicals produced by APPH have been removed outside the cell, thus alleviating the oxidative damage caused by AAPH to human erythrocytes.

## 4. Conclusions

The inhibitory effects of HMP and GHMP on AAPH-induced erythrocytes hemolysis were verified, and the effects of HMP and GHMP on the level of ROS, MDA, and intracellular antioxidant enzymes in RBCs were measured. In the range of 0.025–1.0 mg/mL, HMP and GHMP showed a strong intracellular antioxidant ability in human erythrocytes, and the sample itself showed no damage to red blood cells. HMP possessed stronger antioxidant activity than GHMP. HMP and GHMP significantly attenuated AAPH-induced oxidative hemolysis and normalized the morphological features. Furthermore, HMP and GHMP effectively attenuated AAPH-induced oxidative stress by inhibiting intracellular ROS generation, reducing MDA accumulation, maintaining GSH/GSSG balance, and preserving the activities of SOD, GSH-Px, and CAT. Therefore, both HMP and GHMP possess promising potential as natural antioxidants and can be further explored for application in the development of functional foods.

## Figures and Tables

**Figure 1 foods-12-03468-f001:**
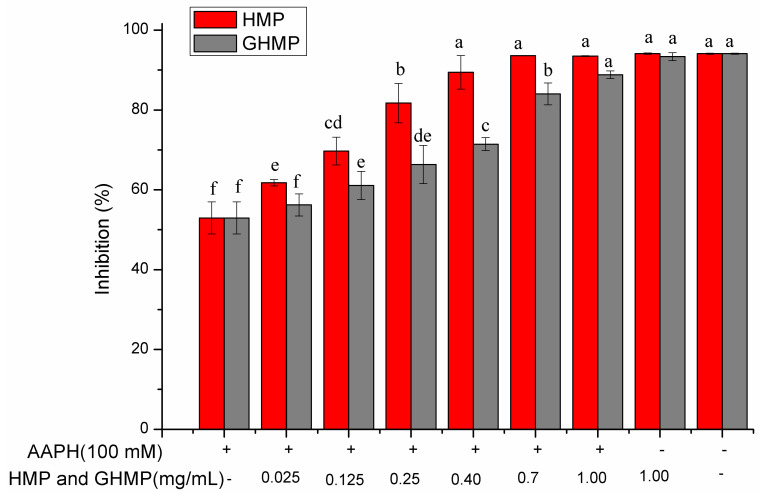
Inhibitory ability of HMP and GHMP on the hemolysis of human erythrocytes at concentrations from 0.025 to 1.0 mg/mL. Different letters at the same detected index refer to significant differences (*p* < 0.05). HMP: neutrase-hydrolysates of mulberry leaf protein; GHMP: in vitro gastrointestinal digests of HMP.

**Figure 2 foods-12-03468-f002:**
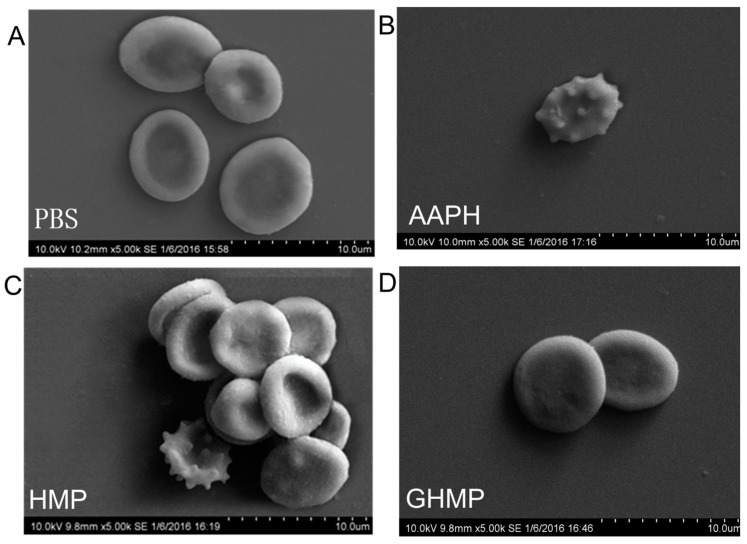
(**A**–**D**) The SEM micrographs of human erythrocyte.

**Figure 3 foods-12-03468-f003:**
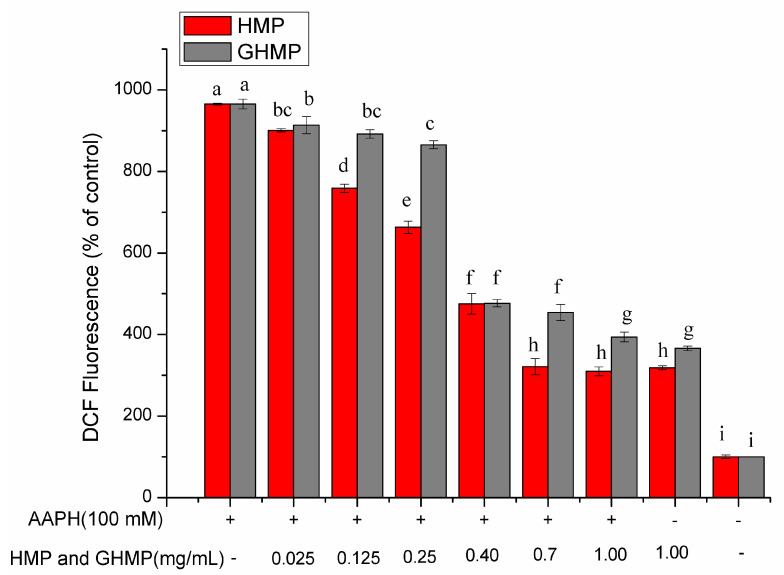
Effects of HMP and GHMP on the generation of AAPH-induced ROS in human erythrocytes at concentrations from 0.025 to 1.0 mg/mL. Different letters at the same detected index refer to significant differences (*p* < 0.05).

**Figure 4 foods-12-03468-f004:**
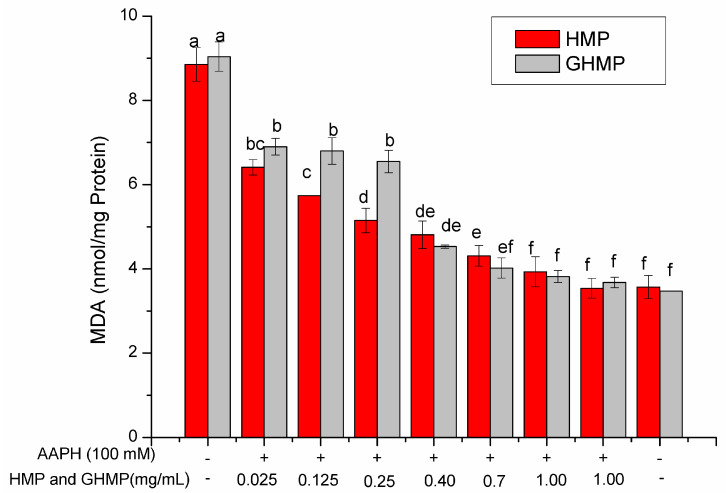
Effects of HMP and GHMP on AAPH-induced MDA accumulation in human erythrocytes at concentrations from 0.025 to 1.0 mg/mL. Different letters at the same detected index refer to significant differences (*p* < 0.05).

**Figure 5 foods-12-03468-f005:**
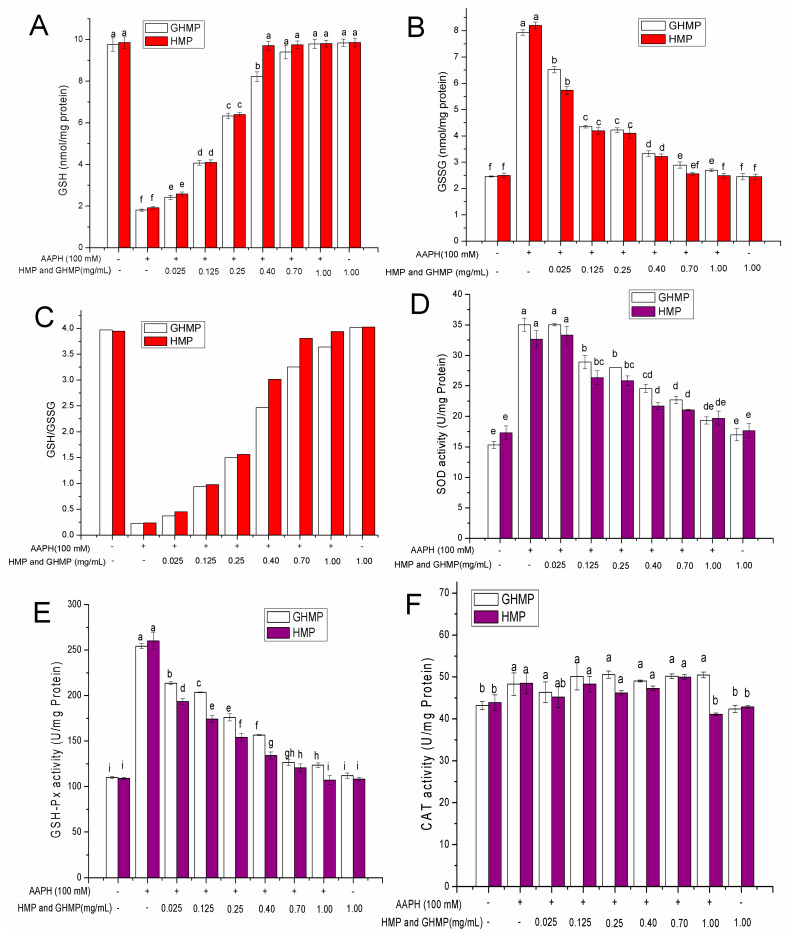
Effects of HMP and GHMP on AAPH-induced changes in GSH (**A**), GSSG (**B**), GSH/GSSG (**C**), SOD (**D**), GSH-Px (**E**), and CAT (**F**) in human erythrocytes at concentrations from 0.025 to 1.0 mg/mL. Different letters at the same detected index refer to significant differences (*p* < 0.05).

**Figure 6 foods-12-03468-f006:**
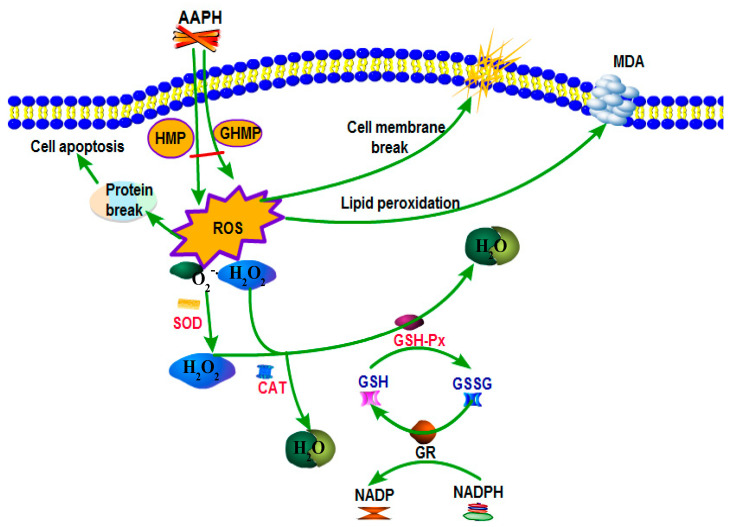
Possible intracellular antioxidant mechanisms of HMP and GHMP attenuate AAPH-induced oxidative damage.

## Data Availability

Data is contained within the article.

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
