# Peer review of "Protective Effects of Mulberry (Morus atropurpurea Roxb.) Leaf Protein Hydrolysates and Their In Vitro Gastrointestinal Digests on AAPH-Induced Oxidative Stress in Human Erythrocytes"

_foods, 2023, doi:10.3390/foods12183468_

Round 1
Reviewer 1 Report
Dear authors,
Let me start by congratulating you on the article. Overall, the manuscript is complex and contain new valuable and interesting information’s. When it comes to the methodology, I really liked how well-organized everything was and how many tests were run. The discussions are presented in a proper manner, with reference to recent literature data.
I found the study to be interesting, but I think there are some negative aspects that should be clarified and corrected.
- Abstract - is much too general. It is necessary to enter some precise information/data concerning the results.
- The Introduction is appropriate but the aim of the work is not clearly established.
- Lines 197 – 199: Inhibitory ability of HMP and GHMP on the haemolysis of human erythrocytes - What does a, ab, bc, ... g mean? Please describe. Additionally, the discussions concerning this subject are too much general.
- I consider that additional statistical explanations are required for all of the graphs presented in figures 1, 3, and 5 (What do the letters a, ab, bc, and g mean?).
Author Response
Dear reviewer, thank you very much for your kind comments and suggestions. We have read them carefully and answered your comments point by point. We hope our responses will meet your kind approval.
- Abstract - is much too general. It is necessary to enter some precise information/data concerning the results.
Response 1: Thanks for your comment. We have revised the abstract and supplemented the precise data regarding the results in the Abstract section. (Lines 27-31 in the revised manuscript)
- The Introduction is appropriate but the aim of the work is not clearly established.
Response 2: Thanks for your advice. The aim of this work is to elucidate the antioxidant mechanism of HMP and GHMP at the cellular level by further validating the inhibitory effects of HMP and GHMP on AAPH-induced erythrocyte hemolysis using human erythrocytes. We have given a clear statement of the aim of this work on Lines 92-98 in the revised manuscript.
- (1) Lines 197 – 199: Inhibitory ability of HMP and GHMP on the haemolysis of human erythrocytes - What does a, ab, bc, ... g mean? Please describe. (2) Additionally, the discussions concerning this subject are too much general.
Response 3: Thank you very much for your suggestions. (1) Different letters (a, b, c, d…) refer to significant differences (p < 0.05). We have added the explanation on Lines 217-218 in the revised version. (2) We have rewritten this part and added some discussions on Lines181-213 in the revised manuscript
- I consider that additional statistical explanations are required for all of the graphs presented in figures 1, 3, and 5 (What do the letters a, ab, bc, and g mean?).
Response 5: Thanks for your comment. Different letters (a, b, c, d…) refer to significant differences (p < 0.05), the same letter on the label indicates no significant difference (p > 0.05). We have added the explanation in the bottom of the figures in the revised manuscript. (Lines 217, 274, 298, 362)

Reviewer 2 Report
In the presented manuscript the authors analyzed the protective effects of mulberry leaf protein hydrolysates on AAPH-induced oxidative stress in human erythrocytes. Moreover, the in vitro gastrointestinal digests was also studied. It was found that both mulberry leaf protein hydrolysates and its in vitro gastrointestinal digests could reduce the AAPH-induced erythrocytes oxidative haemolysis via suppressing the generation of reactive oxygen species level. Consequently, they could be used as natural antioxidants, with the potential for further application of developing the functional foods.
The manuscript is well-written in general and the experiment was well-planned. The results are properly described and discussed. I have only a few comments.
Major
Fig. 6 should be deleted from the conclusion and included in the Results and Discussion section.
Figures 1, 4 and 5. Delete the Figure 1, Figure 4 and Figure 5 on bottom of the charts. It is repeated in the figure description.
Line 125: Please include the condition of freeze-drying and used equipment.
Minor
Literature should be formatted according to the Foods journal requirements.
Author Response
Dear reviewer, thank you very much for your kind comments and suggestions. We have read them carefully and answered your comments point by point. We hope our responses will meet your kind approval.
Major
- 6 should be deleted from the conclusion and included in the Results and Discussion section.
Response 1: Thanks for your advice. We’ve moved Fig. 6 and added the Possible antioxidant mechanism of HMP and GHMP attenuate AAPH-induced oxidative damage to Results and Discussion (on Lines 405-439), and modified Conclusions in the revised version (on Lines 441-451).
- Figures 1, 4 and 5. Delete the Figure 1, Figure 4 and Figure 5 on bottom of the charts. It is repeated in the figure description.
Response 2: Thanks very much for your reminding. We’ve corrected it in the revised version. (Lines 215, 296, 359)
- Line 125: Please include the condition of freeze-drying and used equipment.
Response 3: Thanks for your comment. After desalting, the sample was pre-frozen at -80 °C for 12h and then freeze-dried with a vacuum freeze dryer (SCIENTZ-18N, Ningbo, China) for 36 h to obtain GHMP. We’ve supplemented that on Lines 126-128 in the revised manuscript.
Minor
- Literature should be formatted according to the Foods journal requirements.
Response 4: Thanks very much for your reminding. We’ve noticed that and have corrected the formats regarding to the Foods journal requirements in the revised version.

Reviewer 3 Report
This study investigated the antioxidant activity of HMP and GHMP induced by AAPH. There are major concerns about the experimental method.
As shown in Figure 6, an image of HMP and GHMP were taken into cells and MS was written HMP and GHMP were working in the cells. But in “2.4. Assay for erythrocyte haemolysis induced by AAPH”, AAPH is added while HMP and GHMP remain outside the cell.
The possibility of HMP and GHMP was working extracellular has not been completely abandoned. The authors should clarify this point.
What peptides are included in HMP and GHMP.
What does HMP/GHMP in the figure mean?
The text is HMP and GHMP.
Manuscript has grammatical errors, please check.
Author Response
Dear reviewer, thank you very much for your comments and suggestions. We have read them carefully and answered your comments point by point. We hope our responses will meet your kind approval.
- As shown in Figure 6, an image of HMP and GHMP were taken into cells and MS was written HMP and GHMP were working in the cells. But in “2.4. Assay for erythrocyte haemolysis induced by AAPH”, AAPH is added while HMP and GHMP remain outside the cell. The possibility of HMP and GHMP was working extracellular has not been completely abandoned. The authors should clarify this point.
Response 1: Thanks for your comment. In addition to intracellular antioxidant mechanism, there may also be some extracellular mechanism, because our previous studies have shown that both HMP and GHMP can effectively remove DPPH, ABTS+, and superoxide anions free radicals1-2. In this research, HMP and GHMP were added before APPH, so it is likely that part of the free radicals produced by APPH have been removed outside the cell, thus alleviating the oxidative damage caused by AAPH to human erythrocytes. (Lines 430-439)
- What peptides are included in HMP and GHMP.
Response 2: Thanks for your comments. Our previous research indicated HMP and GHMP were mainly composed of polypeptides (0.5-6.5 kDa) and oligopeptides (<0.5 kDa), respectively (Table 2), and both HMP and GHMP have free radical scavenging activities (DPPH., ABTS+ and O2-)2. Furthermore, three new antioxidant peptides, P1 (SVL, 317 Da), P2 (EAVQ, 445 Da), and P3 (RDY, 452 Da), were purified and obtained from HMP (Table 2, JAFC) 1.
- What does HMP/GHMP in the figure mean? The text is HMP and GHMP.
Response 3: Thanks very much for your comment. The HMP/GHMP in the figure means HMP and GHMP. We have found it is indeed very misleading, so we have made the changes in the revised manuscript. (Lines 215, 272, 296, 359)
References
- Sun, C.; Tang, X.; Ren, Y.; Wang, E.; Shi, L.; Wu, X.; Wu, H., Novel Antioxidant Peptides Purified from Mulberry (Morus atropurpurea Roxb.) Leaf Protein Hydrolysates with Hemolysis Inhibition Ability and Cellular Antioxidant Activity. Journal of Agricultural and Food Chemistry 2019,67(27), 7650-7659.
- Sun, C.; Wu, W.; Yin, Z.; Fan, L.; Ma, Y.; Lai, F.; Wu, H., Effects of simulated gastrointestinal digestion on the physicochemical properties, erythrocyte haemolysis inhibitory ability and chemical antioxidant activity of mulberry leaf protein and its hydrolysates. International Journal of Food Science and Technology 2018,53(2), 282-295.

Round 2
Reviewer 3 Report
The manuscript has been revised well.